# *Colletotrichum gloeosporioides* Swiftly Manipulates the Transcriptional Regulation in *Citrus sinensis* During the Early Infection Stage

**DOI:** 10.3390/jof10110805

**Published:** 2024-11-20

**Authors:** Siyu Zhang, Xinyou Wang, Wei Zeng, Leijian Zhong, Xiaoyong Yuan, Zhigang Ouyang, Ruimin Li

**Affiliations:** 1College of Life Sciences, Gannan Normal University, Ganzhou 341000, China; zsiyu0022@163.com (S.Z.); xinyouwang@gnnu.edu.cn (X.W.); zengwei@gnnu.edu.cn (W.Z.); zhongleijian@gnnu.edu.cn (L.Z.); gzyuanxiaoyong@163.com (X.Y.); 2National Navel Orange Engineering Research Center, Ganzhou 341000, China; 3Jiangxi Provincial Key Laboratory of Pest and Disease Control of Featured Horticultural Plants, Ganzhou 341000, China

**Keywords:** *Citrus sinensis*, *Colletotrichum gloeosporioides*, ROS, flavonoid, transcriptional regulation

## Abstract

*Citrus* spp. represent an economically important fruit tree crop worldwide. However, molecular mechanisms underlying the interaction between citrus and the *Colletotrichum gloeosporioides* remain largely unexplored. In this study, we analyzed the physiological and transcriptomic changes in *Citrus sinensis* at different stages of incubation with *C. gloeosporioides*. The results indicated that *C. gloeosporioides* infection rapidly triggered necrosis in the epicarp of *C. sinensis* fruits, decreased the total flavonoid contents, and suppressed the activity of catalase, peroxidase, and superoxide dismutase enzymes. Upon inoculation with *C. gloeosporioides*, there were 4600 differentially expressed genes (DEGs) with 1754 down-regulated and 2846 up-regulated after six hours, while there were only 580 DEGs with 185 down-regulated and 395 up-regulated between six and twelve-hours post-inoculation. Gene Ontology and the Kyoto Encyclopedia of Genes and Genomes enrichment analysis indicated that the DEGs, which exhibited consistent up-regulation, were associated with metabolic processes and stress responses. Through Weighted Gene Co-Expression Network Analysis, 11 key genes have been identified that could potentially play a role in the transcriptional regulation of this process, including the transcription factor bHLH189. Furthermore, the infection of *C. gloeosporioides* had a notable effect on both the flavonoid metabolism and the metabolic pathways related to reactive oxygen species. Our findings help to understand the interaction between citrus and *C. gloeosporioides* and unveil how new insights into how *C. gloeosporioides* circumvents citrus defense mechanisms.

## 1. Introduction

The citrus industry plays a crucial role in the global agricultural sector, making a significant contribution to economies, especially in developing nations [1,2,3,4]. Oranges, lemons, limes, and grapefruits are indispensable for human nutrition, offering essential nutrients such as Vitamin C [5]. However, the citrus industry is significantly affected by various diseases, such as citrus huanglongbing (HLB), citrus bacterial canker, and citrus anthracnose [6,7,8,9]. The diseases cause significant fruit abscission, diminish fruit characteristics, and result in elevated operational expenses [10,11]. Hence, effective disease control is crucial for the citrus sector, requiring ongoing scientific investigation and innovation.

Anthracnose is a prevalent fungal disease that impacts various horticultural crops globally, predominantly attributable to *Colletotrichum* species [12,13]. This pathogen targets both aerial and subterranean plant components, resulting in a notable reduction in crop productivity and quality [14,15]. Recently, the disease has emerged as a substantial hurdle for numerous horticultural sectors, diminishing the visual appeal and commercial value of fruits [12], vegetables [16], and ornamental plants [17], thereby causing considerable financial repercussions. Various *Colletotrichum* species such as *C. fructicola*, *C. gloeosporioides*, *C. karstii*, *C. siamense*, and *C. theobromicola* have been recognized as pathogens responsible for citrus anthracnose [9,18]. However, the pathological mechanism of citrus anthracnose remains unclear at present.

The interactions between plants and pathogens represent a critical area of study in plant biology, crucial for enhancing agricultural productivity and managing biosecurity threats [19]. Understanding these interactions at molecular and ecological levels enables the development of disease-resistant crops, thus securing food sources for the growing global population [20]. Plant–pathogen interactions at the molecular level encompass intricate signaling cascades [21,22]. Plants detect pathogenic threats via pattern recognition receptors (PRRs), initiating immune responses upon recognition of pathogen-associated molecular patterns (PAMPs) [23,24,25]. Conversely, pathogens employ diverse evasion tactics, such as secreting effector proteins to suppress plant immunity [26,27]. For instance, the effector Cgfl of *C. graminicola* promotes pathogen invasion by inhibiting chitinase activity in maize leaves [28]. CfEC92 from *C. fructicola* functions as a plant immune suppressor during early infection and contributes to pathogen invasion [29]. However, genetic investigations have unveiled a plethora of plant genes linked to pathogen resistance [30,31]. Harnessing these genes through genetic modification and breeding techniques has facilitated the creation of crop varieties with heightened resistance to a range of diseases [31].

Prior research indicates that *Colletotrichum* spp. infection perturbs plant transcriptome expression profiles, thereby impacting cellular biological processes in plants [32]. *C. graminicola* infection significantly induces defense genes associated with the salicylic acid signaling pathway while also inducing the expression of genes related to the jasmonic acid and ethylene signaling pathways [32]. Inoculating both resistant and susceptible strawberry cultivars with *C. gloeosporioides*, followed by transcriptome analysis, unveiled the involvement of genes linked to defense mechanisms, plant–pathogen interactions, and flavonoid biosynthesis pathways in response to *C. gloeosporioides* infection [33]. In response to *C. fructicola* infection, distinctive metabolic pathways within pear leaves undergo modulation, resulting in divergent symptomatology. The accumulation of jasmonic acid (JA), ethylene (ET), and abscisic acid (ABA) in infected leaves adversely affects chlorophyll metabolism and photosynthetic pathways, concurrently enhancing the expression of senescence-related transcription factors and genes. This ultimately leads to leaf yellowing and defoliation [34]. However, the current understanding of the transcriptional response of citrus to *C. gloeosporioides* is constrained by limited research, and the impact of *C. gloeosporioides* on citrus transcriptional expression is not yet elucidated.

Gannanzao navel orange originated from a bud mutation of the Newhall navel orange [35]. Gannanzao is the earliest maturing navel orange variety in China and the first in Jiangxi Province to have independent intellectual property rights. It is the leading choice for refining the navel orange variety structure in the Gannan region. Gannanzao exhibits excellent growth and fruiting capabilities, robust resistance to drought and cold, high yield, and superior quality [35,36]. The fruit features a vibrant orange–yellow hue, high moisture content, and a mildly sweet yet slightly tangy flavor, enhancing its palatability [35]. Notably, Gannanzao navel orange matures over a month earlier than other varieties, typically reaching harvest readiness by late September. However, the peak occurrence of anthracnose in Ganzhou aligns with this ripening period, occurring in September and October.

To reveal the effect of *C. gloeosporioides* infection on the dynamic expression of citrus transcriptome, probe the pathogenic mechanisms of *C. gloeosporioides*, and concurrently screen for susceptibility genes in the genome of *Citrus sinensis*, we conducted an examination of physiological indicators and transcriptome sequencing at the early infection stages of *C. gloeosporioides*. Additionally, the regulatory network of citrus fruit response to *C. gloeosporioides* infection was scrutinized, incorporating a comprehensive exploration of expression trends pertaining to genes associated with ROS and flavonoid metabolic pathways. This study has potential for unveiling novel insights into the interaction between citrus and *C. gloeosporioides* and providing substantial evidence for the further dissection of the pathogenic mechanisms of *C. gloeosporioides*.

## 2. Materials and Methods

### 2.1. Fungal Isolate and Plant Materials

The highly pathogenic strain of *C. gloeosporioides*, ‘GNNU-1’, was identified from *C. sinensis* cv. ‘Newhall’ fruits exhibiting anthracnose symptoms in an orchard located in Xin-feng County, Jiangxi Province, China. The isolated strain has been identified as *C. gloeosporioides* through the verification process of Koch’s postulates (Appendix A). It is important to note that Xin-feng County is a prominent production hub for navel oranges in China and is heavily affected by anthracnose disease. *C. gloeosporioides* ‘GNNU-1’ was cultivated on PDA medium at 28 °C for further experiments.

Healthy six-year-old *C. sinensis* cv ‘Gannanzao’ plants grafted on *Poncirus trifoliate* were cultivated in a well-managed orchard in Xin-feng County, and fully mature fruits of ‘Gannanzao’ were used for subsequent experiments.

### 2.2. Fruit Inoculation with C. gloeosporioides

The detailed inoculation process is outlined as follows: *C. gloeosporioides* was propagated on PDA medium, and conidia were harvested from 7-day-old cultures via rinsing the culture plates with sterile water and filtration using bi-layered cheesecloth. Pathogenicity tests were performed on freshly detached navel oranges utilizing *C. gloeosporioides*, where they were inoculated with a conidial suspension composed of 2 × 10^5^ spores/mL. Each injection site was administered with 200 µL of the conidial suspension using a sterile 1 mL syringe. The inoculated fruits were subsequently enveloped with transparent plastic films to sustain elevated moisture levels. Fruit epicarp samples were gathered at time points of 0-, 6-, 12-, and 24-h post-inoculation.

### 2.3. Total Flavonoid Content Determination

A 0.5 g segment of citrus epicarp was homogenized under low-temperature conditions using a mortar, and then 50 mL of 70% ethanol was added and subjected to ultrasonic extraction for 30 min. After a 10 min centrifugation at 12,000× *g*, the resulting supernatant was utilized for determining the flavonoid levels. Subsequently, 10 mL of the supernatant was combined with 1 mL of 5% sodium nitrite and left to stand for 5 min. This was followed by the addition of 1 mL of 10% aluminum nitrate solution, which was shaken and allowed to stand for 5 min. Upon the addition of 10 mL of 1 M sodium hydroxide solution, the mixture was left to stand for 15 min, and the absorbance was measured at 510 nm. A standard curve was constructed using a rutin standard solution to quantify the flavonoid content present in the samples.

### 2.4. Enzyme Activity Determination

The catalase (CAT) activity was evaluated as previously detailed with minor adjustments [37]. An amount of 0.1 g of citrus epicarp samples was blended with 400 μL of 0.1% cold TCA solution, homogenized to achieve a uniform slurry, and centrifuged at 12,000× *g* at low temperatures for 15 min. Subsequently, 10 μL of the supernatant was combined with 100 μL of 1 M potassium phosphate buffer and 100 μL of 1 M potassium iodide solution and incubated for 5 min. The absorbance at 390 nm was then determined.

To assess peroxidase (POD) activity, 0.5 g of citrus epicarp samples was mixed with 6 mL of phosphate buffer in a mortar and homogenized. The resulting mixture was centrifuged at 8000× *g* for 15 min to obtain the enzyme extraction solution. The reaction mixture contained 50 mL of 100 mM phosphate buffer (pH 6.0) with the addition of 28 μL of guaiacol, which was dissolved through heating and stirring. Following cooling, 19 μL of 30% hydrogen peroxide was added. Subsequently, 3 mL of the reaction mixture was combined with 1 mL of the enzyme extraction solution, and shaken, and the absorbance was promptly measured at 470 nm using a TECAN infinite 200 Pro microplate reader (TECAN, Männedorf, Switzerland). Absorbance readings were taken every minute, and the rate of absorbance change per minute represented one unit of enzyme activity.

To evaluate superoxide dismutase (SOD) activity, 0.5 g of citrus epicarp samples was homogenized with phosphate buffer, following the methods outlined for assessing POD activity. Subsequently, 50 μL of the enzyme extraction solution was added to the reaction mixture (2 mL of 14.5 mM methionine, 0.8 mL of 5 mM Nitroblue tetrazolium, 0.8 mL of 5 mM riboflavin, and 0.8 mL of EDTA). The absorbance was measured at 560 nm using a spectrophotometer to determine enzyme activity. One unit of SOD enzyme activity was defined as the enzyme amount that reduces the absorbance of control by 50%.

### 2.5. RNA-Seq Library Construction and Analysis

Total RNA was extracted and purified from the samples using TRIzol (Thermo Fisher, Waltham, MA, USA) following the manufacturer’s instructions. The quality and purity of the RNA were evaluated using a NanoDrop ND-1000 spectrophotometer (Thermo Fisher, Waltham, MA, USA), and RNA integrity was assessed with a Bioanalyzer 2100 (Agilent, Santa Clara, CA, USA). Samples with an RNA concentration exceeding 50 ng/μL, an RNA Integrity Number (RIN) greater than 7.0, and a total RNA amount over 1 μg were deemed suitable for further analysis. mRNA was isolated using two rounds of purification with Oligo (dT) magnetic beads (Thermo Fisher, USA) to selectively bind mRNA with PolyA tails. The mRNA was then fragmented at 94 °C for 5–7 min using the NEBNext Magnesium RNA Fragmentation Module (NEB, Ipswich, MA, USA). This fragmented RNA was reverse-transcribed into cDNA using Invitrogen SuperScript II Reverse Transcriptase (Invitrogen, Carlsbad, CA, USA). Second-strand cDNA synthesis was performed using *E. coli* DNA Polymerase I (NEB, USA) and RNase H (NEB, USA), incorporating dUTP Solution (Thermo Fisher, USA) to blunt the DNA ends, followed by the addition of a single A base to facilitate adapter ligation. The double-stranded DNA was size-selected, purified using magnetic beads, and treated with UDG enzyme (NEB, USA) to prepare the library. The library underwent PCR with a cycle profile of pre-denaturation at 95 °C for 3 min, denaturation at 98 °C for 15 s, annealing at 60 °C for 15 s, and extension at 72 °C for 30 s for 8 cycles, with a final extension at 72 °C for 5 min. This library, with an average fragment size of 300 bp ± 50 bp, was sequenced on an BGISEQ-500 (BGI tech, Shenzhen, China) in a paired-end 150 bp mode.

The raw sequencing data underwent quality assessment using FastQC (https://www.bioinformatics.babraham.ac.uk/projects/fastqc/ (accessed on 13 October 2023)) and adapter trimming via Trimmomatic [38], after which the resulting sequencing data were utilized for subsequent analysis. Gene expression levels in different samples were determined using Kallisto v1.0 [39] with a defined Kmer size of 31, and the *C. sinensis* genome v3.0 was used as the reference genome [40]. Subsequently, differentially expressed genes (DEGs) between samples were identified through DESeq2 [41] with criteria of |log2FC| > 1 and FDR < 0.05. The heatmaps were created with Pheatmap v1.0.12. The Venn diagram was visualized using the interactiVenn software v1.0 [42]. Gene clustering trend analysis was conducted using Mfuzz v3.20 [43].

### 2.6. Gene Functional Annotation and Enrichment

All genes within the genome of *C. sinensis* were subjected to annotation using eggNOG-mapper 2.1.12, with searches executed against the eggNOG 5 database under default search filter parameters [44]. The annotation results included Gene Ontology (GO) and Kyoto Encyclopedia of Genes and Genomes (KEGG) annotations for individual genes, which were subsequently utilized for GO and KEGG enrichment analyses. Moreover, gene annotation results were validated through NCBI BLAST [45] and SMART annotation [46] processes.

### 2.7. Weighted Gene Co-Expression Network Analysis (WGCNA)

The WGCNA analysis was conducted using WGCNA-shinyApp v1.0 (https://github.com/ShawnWx2019/WGCNA-shinyApp (accessed on 29 November 2023)) [47]. Raw gene count values underwent normalization via the variance-stabilizing transformation method [48] and were subsequently subjected to two rounds of gene-set filtering. Initially, genes with less than 10 count values in 90% of samples were excluded. Subsequently, genes were further refined using the ‘median absolute deviation’ method [49]. The normalized count values of the retained genes were employed to determine the appropriate power value. A module network was then constructed with parameters set at ‘min Module size = 30’ and ‘module cuttree height = 0.25’. The correlation between modules and trait data was calculated, and significant ‘module-trait’ associations were utilized to identify hub genes.

### 2.8. Analysis of the ROS-Related Metabolism and Flavonoid Metabolism Pathway

Specifically, genes involved in reactive oxygen species (ROS) metabolism and the flavonoid metabolism pathway underwent sequence alignment with annotated genes in *Arabidopsis thaliana* to validate their functions [50]. Genes annotated as phenylalanine ammonia lyase (*PAL*), cinnamate-4-hydroxylase (*C4H*), 4-coumarate:CoA ligase (*4CL*), chalcone synthase (*CHS*), chalcone isomerase (*CHI*), and flavanone 3-hydroxylase (*F3H*) associated with the flavonoid metabolism pathway, along with NADPH oxidase (*RBOH*), superoxide dismutase (*SOD*), catalase (*CAT*), peroxiredoxin (*PrxR*), thioredoxin (*Trx*), ascorbate peroxidase (*APX*), monodehydroascorbate reductase (*MDAR*), dehydroascorbate reductase (*DHAR*), and glutaredoxin (*GLR*) related to ROS metabolism [51], were identified from the DEGs. The expression profiles of these genes were further analyzed and visualized using Pheatmap v1.0.12 [52].

### 2.9. Statistic Analysis

Statistical analyses were conducted using SPSS v19.0. The Duncan test was used to determine the significance of differences between groups, with *p*-values < 0.05 considered statistically significant.

## 3. Results

### 3.1. C. gloeosporioides Infection Swiftly Induces Necrosis of C. sinensis Fruits

Following incubation, infection by *C. gloeosporioides* rapidly triggers necrosis in *C. sinensis* fruits (Figure 1 and Appendix A). In comparison to the control samples (Figure 1A), necrosis in the inoculation regions started at 6 h after inoculation (Figure 1B). There was a noticeable increase in necrosis by 12 h after inoculation (Figure 1C), and, by 24 h after inoculation, it had progressed to a severe level (Figure 1D). We isolated the total RNA from the aforementioned samples. Due to the severe symptom in the fruit, the tissue appeared water-soaked, and we observed that the RNA from the samples at 24 h post *C. gloeosporioides* infection was highly degraded. As a result, we used control samples and samples of 6- and 12 h post *C. gloeosporioides* infection for further experiments.

### 3.2. C. gloeosporioides Infection Reduced the Flavonoid Contents and Disturbed the Enzyme Activities of SOD, POD, and CAT in C. sinensis Fruits

There was a significant decrease in total flavonoids in the epicarp of *C. sinensis* fruits after *C. gloeosporioides* infection when compared to the control samples, as measured at 6- and 12 h post-infection (Figure 2A). Interestingly, *C. gloeosporioides* infection also significantly inhibited the catalytic activity of SOD, POD, and CAT in the epicarp of *C. sinensis* fruits (Figure 2B–D).

### 3.3. DEGs Induced by C. gloeosporioides Infection Primarily Occurs in the Early Stages

In comparison to the control samples, the inoculation of *C. gloeosporioides* for a duration of 6 h resulted in a total of 4600 DEGs, comprised of 2846 up-regulated and 1754 down-regulated genes (Figure 3A). Extending the inoculation period to 12 h under the same conditions yielded 5253 DEGs, with 2757 genes up-regulated and 2496 genes down-regulated (Figure 3B). A comparative analysis of the RNA-seq data at 12 h versus 6 h post-inoculation revealed a mere 580 DEGs, consisting of 395 up-regulated and 185 down-regulated genes (Figure 3A). Interestingly, a 12 h inoculation period with *C. gloeosporioides* elicited more down-regulated gene expressions than a 6-h duration (Figure 3A). Notably, despite these fluctuations, the DEGs lacked any significant chromosomal distribution pattern, thus negating the possibility of chromosome-specific distribution trends (Figure 3B–D).

### 3.4. The Up-Regulation DEGs Mainly Involved in Defense Response and Metabolic Process

Upon the analysis of the differential gene sets between samples at 6 h post-inoculation of *C. gloeosporioides* and control (Cg6h vs. Cg0h), at 12 h post-inoculation of *C. gloeosporioides* and control (Cg12h vs. Cg0h), and between 12- and 6-h post-inoculation of *C. gloeosporioides* (Cg12h vs. Cg6h), it was ascertained that 186 DEGs were consistent across all three comparisons (Figure 4A). The most extensive overlap of common DEGs occurred in ‘Cg6h vs. Cg0h’ and ‘Cg12h vs. Cg0h’, amassing a total of 2908 genes (Figure 4A). While ‘Cg6h vs. Cg0h’ and ‘Cg12h vs. Cg0h’ shared a significant number of DEGs, each gene set still hosted exclusive DEGs, such as the 1892 DEGs specific to ‘Cg12h vs. Cg0h’ (Figure 4A). After eliminating duplicate entries, a sum total of 6810 unique genes were delineated. A heatmap analysis of these DEGs revealed two expression categories (Figure 4B). A more detailed exploration of expression trends, facilitated by Mfuzz, disclosed six distinct trends, labeled clusters 1–6 (Figure 4C). Genes in cluster 1 demonstrated mild up-regulation after 6 h of inoculation, followed by significant up-regulation after 12 h of *C. gloeosporioides* (Figure 4C). Genes in cluster 2 exhibited consistent down-regulation (Figure 4C), whereas genes in cluster 3 and 4 initially up-regulated and then significantly and slightly down-regulated, respectively (Figure 4C). Genes in cluster 5 initially down-regulated significantly and then slightly, and genes in cluster 6 remained consistently up-regulated (Figure 4C). The GO enrichment analysis of the six gene clusters revealed that genes in cluster 1 were predominantly associated with ‘metabolic process’ and ‘response to stress’ (Figure 4D), while genes in cluster 6 were primarily tied to ‘response to stress’ (Figure 4E).

The KEGG enrichment analysis demonstrated that the genes in cluster 1 participated in various metabolic pathways, predominantly those linked to the biosynthesis of secondary metabolites. These pathways included ‘anthocyanin biosynthesis’, ‘phenylpropanoid biosynthesis’, ‘tryptophan metabolism’, and ‘glucosinolate biosynthesis’ (Figure 5A). In addition, genes in cluster 6 mainly correlated with metabolic pathways such as ‘Glycolysis/Gluconeogenesis’, the ‘Citrate Cycle’, ‘biosynthesis of secondary metabolites’, and ‘Galactose metabolism’ (Figure 5B).

### 3.5. WGCNA Revealed Key Modules and Genes Involved in Transcriptional Regulation

In order to further explore the hub genes involved in transcriptional regulation in *C. sinensis* during the inoculation of *C. gloeosporioides*, we utilized WGCNA to construct co-expression modules (Figure 6A). Following the initial filtering process, which excluded genes with less than 10 count values in 90% of samples, a second refinement was carried out using the ‘median absolute deviation’ method. As a result, a total of 13,805 genes were selected for WGCNA (Figure 6A and Appendix A). By correlating the co-expression modules with different time points of *C. gloeosporioides* inoculation, we discovered two gene modules significantly associated with ‘Cg6h’ (Figure 6B). Specifically, the green module showed a significant negative correlation with Cg6h, while the red module showed a significant positive correlation with Cg6h (Figure 6B). Similarly, two gene modules were found to be significantly associated with ‘Cg12h’, with the blue module and grey module both exhibiting significant negative correlations with ‘Cg12h’ (Figure 6B). Further analysis of the correlation between significance genes and module membership in the four modules revealed a significant correlation between significance genes and module membership in each module (Figure 6C–F).

Further investigation into hub genes within the four modules and the construction of co-expression networks indicated that the core genes in the green module were annotated as ‘Cytochrome b-c1 complex subunit Rieske’ and ‘ATP synthase subunit Epsilon’ (Figure 7A, Appendix A). In the red module, the core genes were identified as ‘Splicing factor U2af large subunit A’, ‘ADP, ATP carrier protein 3’, and an ‘Uncharacterized protein’ (Figure 7B, Appendix A). In the blue module, the core gene was denoted as the ‘F-box/LRR-repeat protein’ (Figure 7C, Appendix A), and, in the grey module, the core genes were comprised of ‘Leucine zipper-EF-hand-containing transmembrane protein 1’, ‘Nuclear transcription factor Y subunit A-3’, ‘Transcription factor bHLH169’, and two uncharacterized proteins (Figure 7D, Appendix A).

### 3.6. C. gloeosporioides Infection Disturbed the Flavonoid Metabolism and ROS-Related Metabolism Pathway

To elucidate the impact of *C. gloeosporioides* infection on the flavonoid metabolic pathway in *C. sinensis*, we scrutinized DEGs associated with flavonoid metabolism. A total of 14 DEGs engaged in flavonoid metabolism, with an up-regulation of eight genes and a down-regulation of six genes, were identified (Figure 8). Specifically, three *PAL* and four *4CL* genes were up-regulated, while one *4CL* gene was down-regulated. One *CHS* gene was up-regulated, while another *CHS* gene and one *CHI* gene experienced down-regulation (Figure 8). Notably, three *F3H* genes also showed down-regulation. Despite the larger quantity of up-regulated genes tied to flavonoid metabolism, pivotal genes in flavonoid synthesis, such as *CHI* and *F3H*, underwent down-regulation (Figure 8). As the upstream genes that catalyze 4-coumaroyl CoA production also serves as a substrate for lignin biosynthesis, the augmented expression of upstream genes for flavonoid synthesis due to *C. gloeosporioides* infection may result in flavonoid synthesis inhibition.

Similarly, we identified 31 DEGs related to ROS metabolism, with 16 genes up-regulated and 15 genes down-regulated (Figure 9). Specifically, four *RBOH* genes experienced up-regulation, and one *RBOH* gene saw down-regulation (Figure 9). One *SOD* gene was up-regulated, with two *SOD* genes seeing down-regulation (Figure 9). Moreover, one *CAT* gene and three *PrxR* genes were down-regulated, while six *Trx* genes were up-regulated and four *Trx* genes down-regulated (Figure 9). One *APX* gene was up-regulated, and another *APX* gene, one *MDAR* gene, and two *DHAR* genes were down-regulated (Figure 9). Lastly, four *GLR* genes were up-regulated (Figure 9).

## 4. Discussion

In the present study, we elucidated that *C. gloeosporioides* infection has significant impact on the physiological and transcriptomic changes in *C. sinensis*. Our findings demonstrated a decline in total flavonoid content and a decrease in the activities of CAT, POD, and SOD enzymes of *C. sinensis* during *C. gloeosporioides* infection (Figure 10). Moreover, we discovered significant alterations in gene expression, particularly in ROS and flavonoid metabolic pathways. The changes extended to genes integral to metabolic processes, notably those involved in the biosynthesis of secondary metabolites and glycolysis/gluconeogenesis, as well as genes related to defensive responses (Figure 10).

### 4.1. C. gloeosporioides Infection Alters the Dynamic Changes in ROS Metabolism

ROS serve a crucial role as cellular signaling molecules in plant growth, development, and adaptation to environmental variations [53]. Plant resistance mechanisms and pathogen infiltration strategies can be better understood by examining the intricate dynamics of plant–pathogen interactions, with ROS playing a vital role in dissecting these complexities [54,55]. Plant–pathogen infections can disturb the equilibrium of ROS metabolism, leading to harmful bursts of ROS. Despite the potential damage to cellular components, these bursts can also trigger defense mechanisms [56]. For instance, when infected with *Magnaporthe oryzae*, rice triggers the production of ROS in order to limit the expansion of invasive hyphae [55]. In this study, the expression profiles of genes involved in ROS metabolism were found to be significantly changed during *C. gloeosporioides* infection. The *RBOH* gene, which is important for ROS production, was up-regulated, while the expression of enzymes responsible for ROS clearance showed varying patterns, with some being up-regulated and others down-regulated. The key enzyme for clearing hydrogen peroxide, CAT, was down-regulated. Enzymatic assays also revealed a decrease in the activities of SOD, CAT, and POD enzymes. These findings suggest that *C. gloeosporioides* infection triggers ROS production, hinders ROS clearance, and ultimately results in the accumulation of ROS.

ROS, predominantly generated in chloroplasts, mitochondria, and peroxisomes, may exhibit direct antimicrobial properties and function as secondary messengers in defense signal transduction [57,58]. Pathogen infiltration triggers a disruption in electron flow in the plant cell, resulting in ROS overproduction [58]. Then, ROS burst can instigate programmed cell death and restrict the proliferation of biotrophic and hemi-biotrophic pathogens that rely on living host tissue for nutrient sourcing [59,60]. However, some pathogens have developed mechanisms to manipulate ROS signaling to support their colonization, such as the effector proteins produced by *Pseudomonas syringae*, which could regulate host ROS level [61]. The findings of this study indicate the potential for further examination of the effector proteins of *C. gloeosporioides*, identification of effector proteins that may influence ROS levels in citrus, and a detailed investigation into the interaction mechanisms between *Citrus* spp. and *C. gloeosporioides*. Furthermore, the production of ROS induced by pathogens also stimulates the synthesis of secondary metabolites such as flavonoids, carotenoids, and vitamins in hosts, serving as antioxidants, to further mitigate oxidative stress [62,63,64].

### 4.2. C. gloeosporioides Infection Promotes the Degradation of Flavonoids

Flavonoids are a diverse group of plant secondary metabolites that play various roles in plant physiology and interactions with the environment [65,66]. Flavonoids are important for plant defense against pathogens, both through direct antimicrobial properties and by modulating plant immune responses [67,68]. When plants are infected with pathogens, they often increase the production of specific flavonoids [69]. These compounds, such as kaempferol and quercetin, have antimicrobial properties that can inhibit pathogen growth and prevent infections [70,71]. Flavonoids also contribute to physical defense against pathogens by reinforcing cell walls, making it harder for pathogens to invade [72]. Nevertheless, certain pathogens have the ability to bypass flavonoid defenses, either by degrading defensive flavonoids or by developing resistance to flavonoid phytoalexins that aid in their survival and colonization [73,74]. In the initial phases of *C. gloeosporioides* infection, there was a notable decrease in the total flavonoid content within the infected tissue. Given the brief period of inoculation, the likelihood of cells converting total flavonoids into alternative substances is minimal. Hence, it is hypothesized that *C. gloeosporioides* infection enhances its own invasion by breaking down the total flavonoids present in citrus fruits. Moreover, the infection of *C. gloeosporioides* led to a decrease in the expression levels of genes involved in flavonoid synthesis. Therefore, we speculate that *C. gloeosporioides* promotes its rapid colonization by degrading flavonoids while simultaneously inhibiting the synthesis of flavonoids.

### 4.3. C. gloeosporioides Infection Instigates the Up-Regulation of Genes Associated with Metabolic Process and Defensive Response

Plants have developed complex mechanisms to defend against pathogen attacks, which involve the activation of specific genes related to metabolic processes and defense responses [30,75,76]. In the presence of infection, genes that play a role in the synthesis of secondary metabolites such as terpenoids, alkaloids, and flavonoids are often up-regulated [77,78]. In addition to metabolic genes, genes associated with defense responses of plants are also activated. These include pathogenesis-related (PR) genes, which produce proteins with antimicrobial properties, as well as genes involved in the plant hormones metabolic pathways, which play key roles in signaling for plant defense [79,80,81,82]. In our study, we observed an inducement in the expression of genes related to metabolic processes and defense responses in citrus fruit tissues during infection with *C. gloeosporioides*. Enrichment analysis revealed that these up-regulated genes were involved in primary metabolism, secondary metabolism, and defense responses. Despite the rapid expression of these genes, they were unable to effectively suppress *C. gloeosporioides* infection. This suggests that the activation of these genes may be a common response in plants to pathogen infection and that *C. gloeosporioides* possesses virulence factors that allow it to evade citrus defense. Moving forward, our research will focus on identifying the genes responsible for encoding key virulence factors in the *C. gloeosporioides* genome.

### 4.4. Core Genes Involved in Transcriptional Regulation During C. gloeosporioides Infection

WGCNA is a powerful tool for identifying key genes involved in transcriptional regulation during the interaction between plants and pathogens. In this research, we pinpointed central genes in modules that show both positive and negative correlations with the infection of *C. gloeosporioides*. Recent research revealed that the splicing factor GauU2AF35B facilitated the exon splicing of *BAK1*, leading to heightened cotton resistance against *Verticillium wilt* [83]. During the host infection, the gene encoding the ADP/ATP carrier protein in *Peronospora* spp. showed high levels of expression [84]. In our research, we observed a strong correlation between *C. gloeosporioides* infection and the expression of U2AF and ADP/ATP carrier protein 3 genes. This suggests that *C. gloeosporioides* infection may alter alternative splicing events within citrus cells, a process that appears to be closely linked to ATP. The expression levels of *NtLRR1* and *NtLRR2* were significantly elevated under pathogenic incursion [85]; however, our study revealed a negative correlation between a gene encoding an F-box/LRR-repeat protein during infection by *C. gloeosporioides*. F-box/LRR repeat proteins are known for their capacity to enhance plant resistance against pathogens [86,87]. Therefore, *C. gloeosporioides* might manipulate the expression of citrus resistance genes in some way to facilitate its own rapid infection. Similarly, a negative association was observed in the nuclear transcription factor Y subunit A (NF-YA) during *C. gloeosporioides* infection. Prior research has demonstrated that the overexpression of a specific rice *NF-YA* gene (*OsHAP2E*) instigated resistance to *Magnaporthe oryzae* and *Xanthomonas oryzae* pv. Oryzae [88]. The resistance of poplar to *Alternaria alternata*, mediated by the overexpression of *PdbNF-YA11*, was associated with the activation of the JA pathway [89]. The down-regulation of the genes that were responsible for the disease resistance in citrus could be a tactic used by *C. gloeosporioides* to effectively infect citrus plants.

## 5. Conclusions

In conclusion, the present study analyzed the complex interaction between *C. sinensis* and *C. gloeosporioides*, the causal agent of citrus anthracnose. Our research elucidates marked alterations in the transcriptomic profile of *C. sinensis*, particularly within genes pertaining to ROS and flavonoid metabolic pathways, during *C. gloeosporioides* infection. Concurrently, we discovered a multitude of genes actively involved in the metabolic proceedings and defensive responses of *C. sinensis*. Interestingly, we also identified various core genes from the co-expression networks that correlate with *C. gloeosporioides* infection. These insights offer valuable groundwork for breeding disease-resistant citrus varieties and devising effective disease management strategies. Nevertheless, future studies are necessary to validate the functions of these identified genes, using techniques such as gene knock-out and overexpression experiments.

## Figures and Tables

**Figure 1 jof-10-00805-f001:**
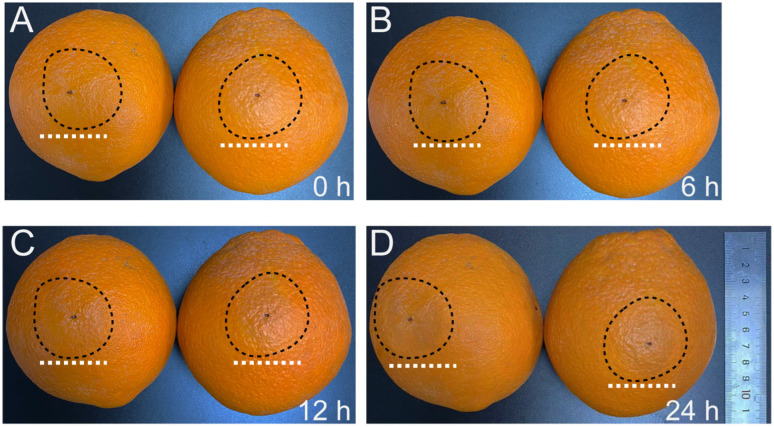
The fruit phenotype changes of *Citrus sinensis* after inoculation with *Colletotrichum gloeosporioides* at different stages. (**A**) Control samples. (**B**) Samples of 6 h post-inoculation. (**C**) Samples of 12 h post-inoculation. (**D**) Samples of 24 h post-inoculation. The white dashed lines and black dashed circles indicate the inoculation regions.

**Figure 2 jof-10-00805-f002:**
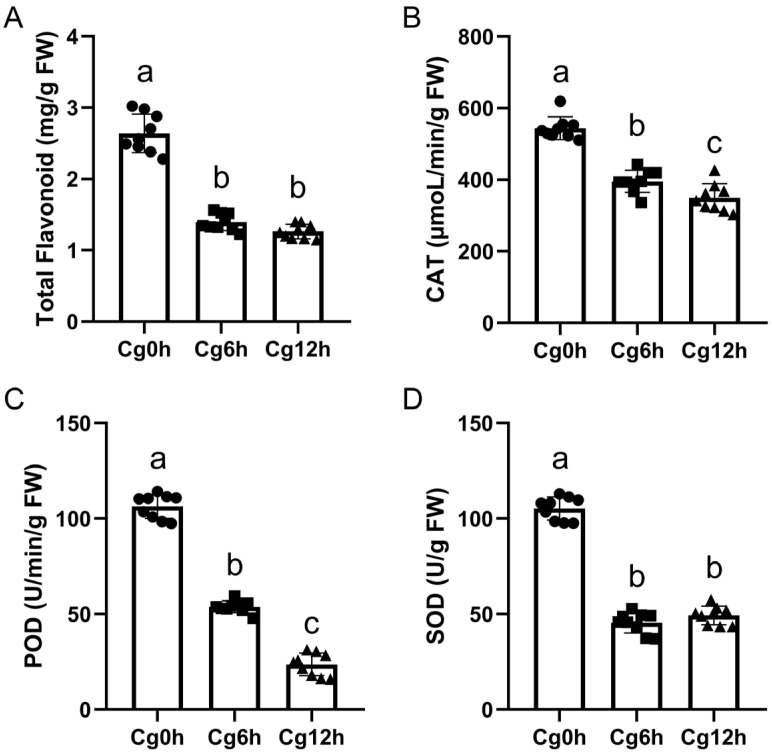
The alterations in physiological indicators of *Citrus sinensis* fruit after inoculation with *Colletotrichum gloeosporioides* at different stages. (**A**) Content of total flavonoid. (**B**) Catalase (CAT) activity. (**C**) Peroxidase (POD) activity. (**D**) Superoxide dismutase (SOD) activity. The letter ‘a’ ‘b’ ‘c’ on the bar denote a statistically significant discrepancy with a *p*-value inferior to 0.05.

**Figure 3 jof-10-00805-f003:**
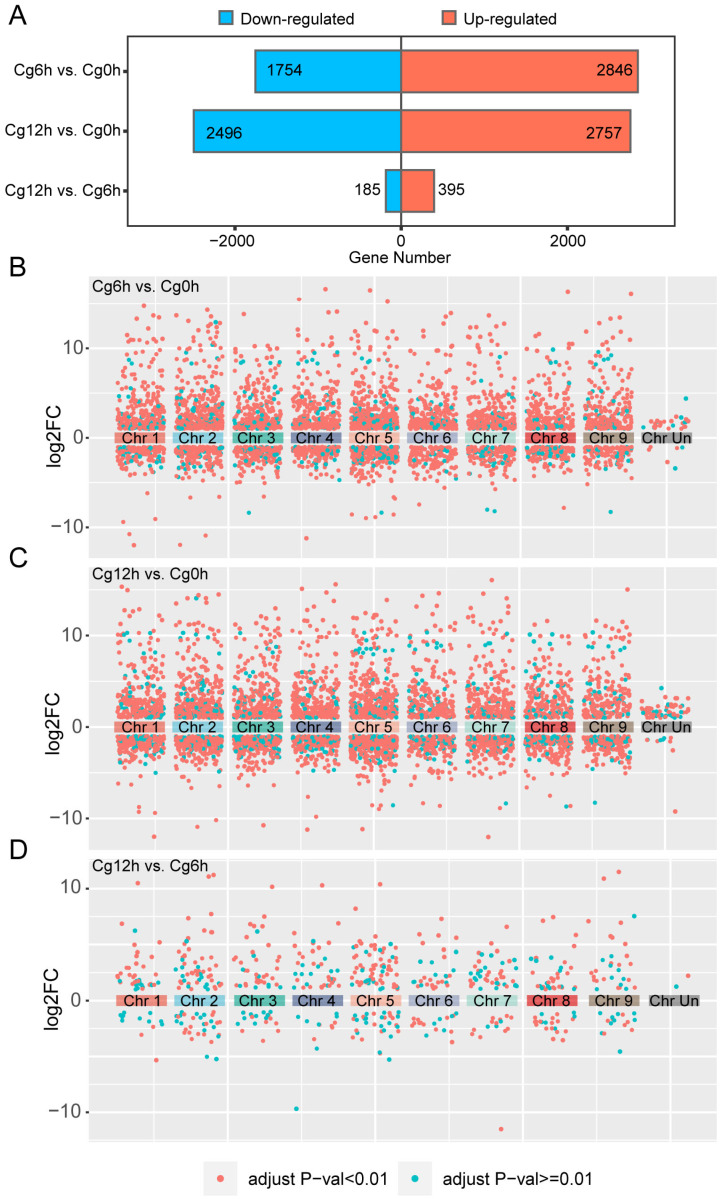
Analysis of differentially expressed genes (DEGs) and their distribution on chromosomes of *Citrus sinensis* during *Colletotrichum gloeosporioides* infection. (**A**) DEGs in *C. sinensis* fruits at various incubation stages of *C. gloeosporioides*. (**B**) Distribution of DEGs between 6 h post-inoculation and control samples on each chromosome of *C. sinensis*. (**C**) Distribution of DEGs between 12 h post-inoculation and control samples on each chromosome of *C. sinensis*. (**D**) Distribution of DEGs between 12 h and 6 h post-inoculation on each chromosome of *C. sinensis*. Cg0h, Cg6h, and Cg12h refer to control samples and samples of 6- and 12 h post-inoculation with *C. gloeosporioides*, respectively. Log2FC represents the log2 of the fold change in the expression value.

**Figure 4 jof-10-00805-f004:**
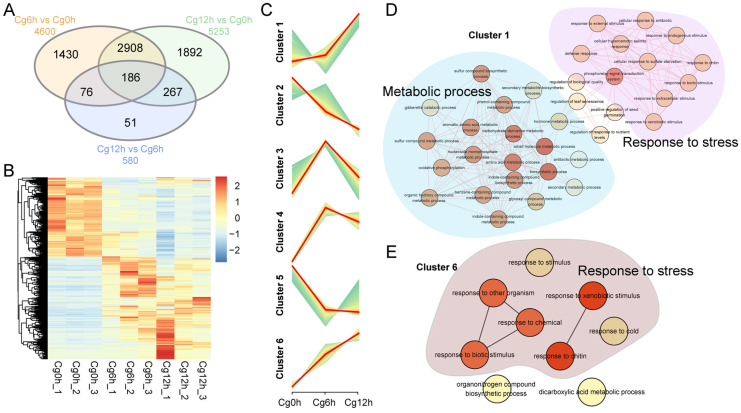
Cluster and GO enrichment analysis of DEGs of *Citrus sinensis* during *Colletotrichum gloeosporioides* infection. (**A**) Venn diagram of DEGs across ‘Cg6h vs. Cg0h’, ‘Cg12h vs. Cg0h’, and ‘Cg12h vs. Cg6h’. (**B**) Expression profiles of DEGs. (**C**) Cluster analysis of DEGs. (**D**) GO enrichment analysis of DEGs in cluster 1. (**E**) GO enrichment analysis of DEGs in cluster 6.

**Figure 5 jof-10-00805-f005:**
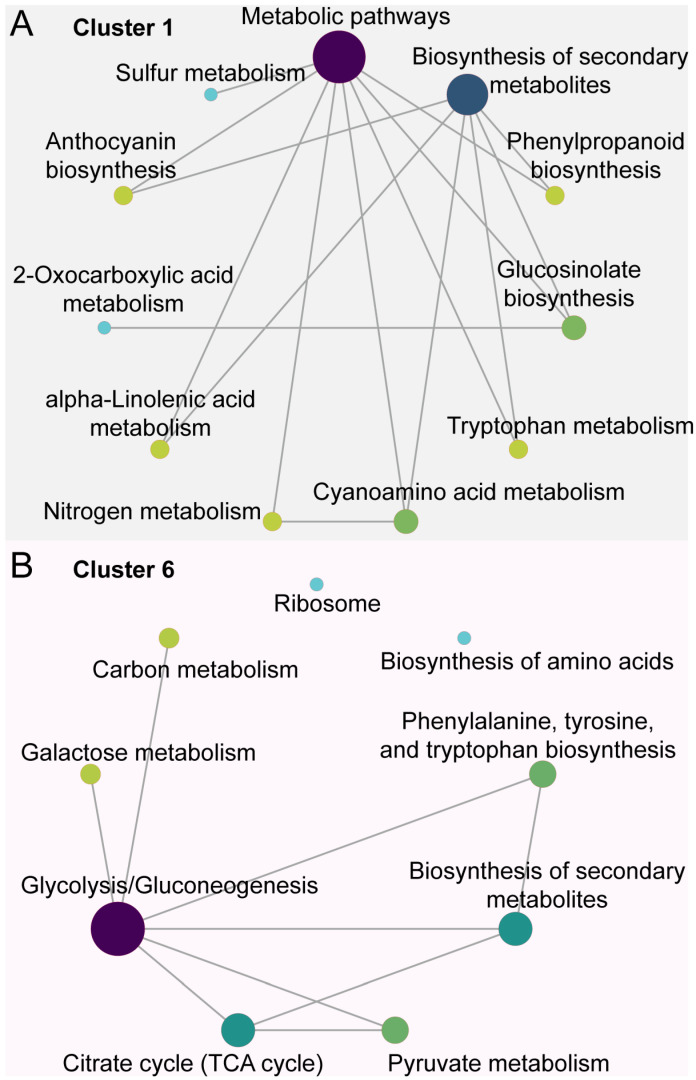
KEGG enrichment analysis of up-regulated DEGs of *Citrus sinensis* during *Colletotrichum gloeosporioides* infection. (**A**) KEGG enrichment analysis of DEGs in cluster 1. (**B**) KEGG enrichment analysis of DEGs in cluster 6.

**Figure 6 jof-10-00805-f006:**
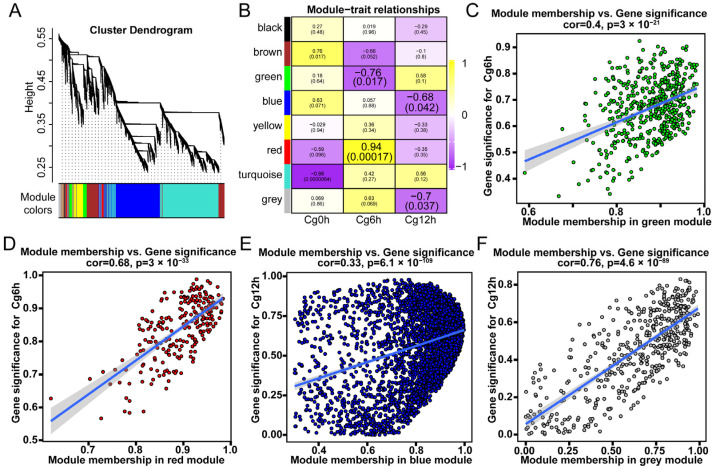
Identifying key modules of gene sets from *Citrus sinensis* associated with *Colletotrichum gloeosporioides* infection by WGCNA. (**A**) Cluster genes into various modules. (**B**) The correlation analysis between modules and incubation stages of *C. gloeosporioides*. The numbers in the rectangular columns show the correlation coefficient and *p* value. The enlarged labels indicate significant modules. The correlation analysis between module membership and gene significance in (**C**) green module, (**D**) red module, (**E**) blue module, and (**F**) grey module.

**Figure 7 jof-10-00805-f007:**
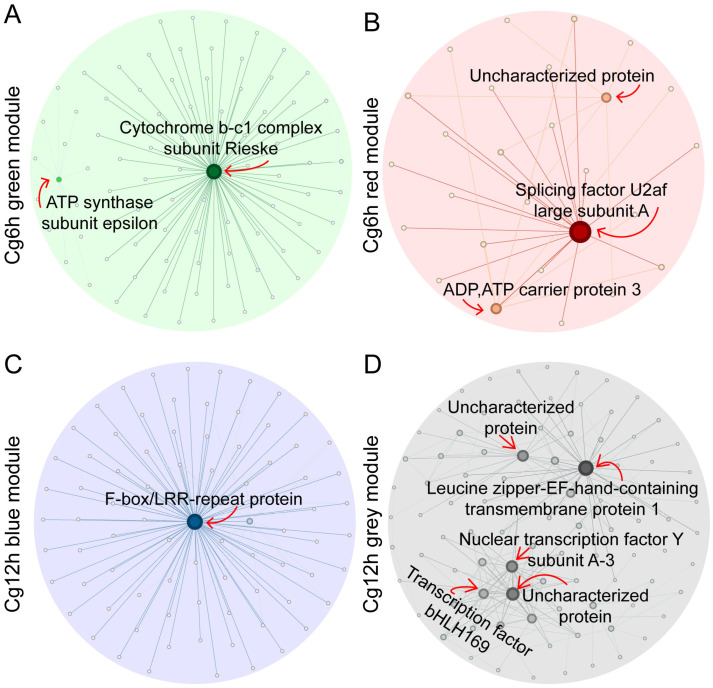
Co-expression networks of each significant module. (**A**) Green module. (**B**) Red module. (**C**) Blue module. (**D**) Grey module. Cg6h and Cg12h indicate samples of 6- and 12 h post-inoculation with *C. gloeosporioides*, respectively. The core genes were annotated in the networks.

**Figure 8 jof-10-00805-f008:**
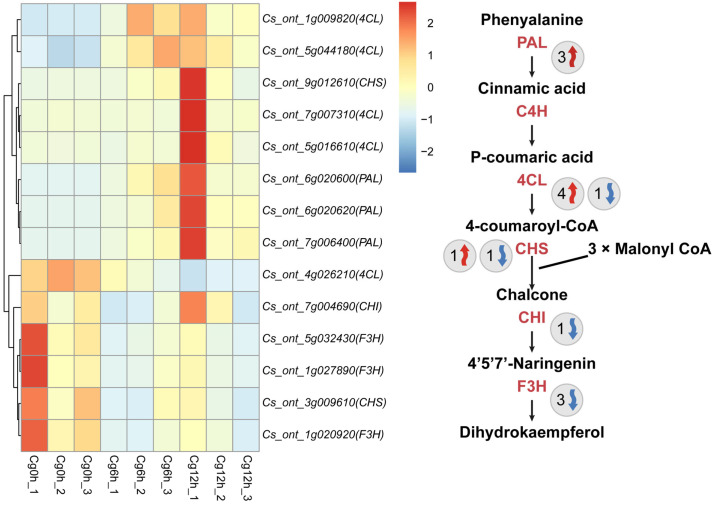
Expression profiles of differential expressed flavonoid metabolism-associated genes of *Citrus sinensis* during *Colletotrichum gloeosporioides* infection. Cg0h, Cg6h, and Cg12h refer to control samples and samples of 6- and 12 h post-inoculation with *C. gloeosporioides*, respectively. The numbers following Cg0h, Cg6h, and Cg12h indicate different biological replicates. Red arrows indicate genes that are up-regulated, while blue arrows represent genes that are down-regulated. The numbers beside the arrows indicate the number of DEGs. PAL, phenylalanine ammonia lyase. C4H, cinnamate-4-hydroxylase. 4CL, 4-coumarate:CoA ligase. CHS, chalcone synthase. CHI, chalcone isomerase. F3H, flavanone 3-hydroxylase.

**Figure 9 jof-10-00805-f009:**
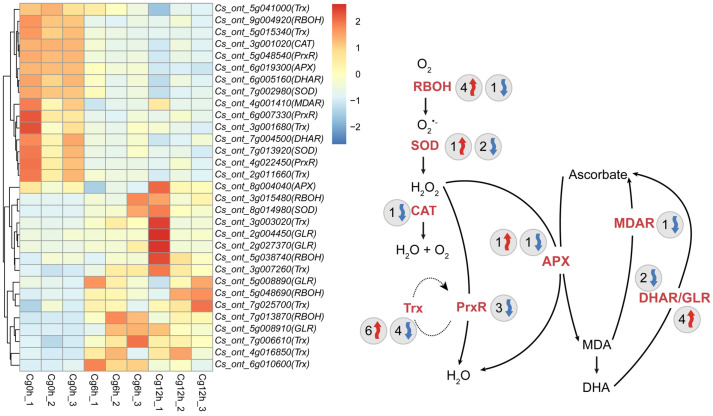
Expression profiles of differential expressed ROS metabolism-associated genes of *Citrus sinensis* during *Colletotrichum gloeosporioides* infection. Cg0h, Cg6h, and Cg12h refer to control samples and samples of 6- and 12 h post-inoculation with *C. gloeosporioides*, respectively. The numbers following Cg0h, Cg6h, and Cg12h indicate different biological replicates. Red arrows indicate genes that are up-regulated, while blue arrows represent genes that are down-regulated. The numbers beside the arrows indicate the number of DEGs. RBOH, NADPH oxidase. SOD, superoxide dismutase. CAT, catalase. PrxR, peroxiredoxin. Trx, thioredoxin. APX, ascorbate peroxidase. MDAR, monodehydroascorbate reductase. DHAR, dehydroascorbate reductase. GLR, glutaredoxin.

**Figure 10 jof-10-00805-f010:**
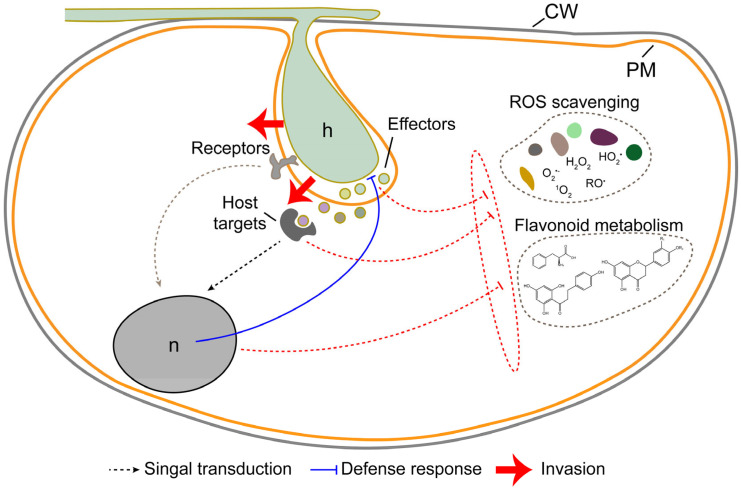
Modules of *Colletotrichum gloeosporioides* swiftly manipulate the transcriptional regulation in *Citrus sinensis* during the early infection stage. The infection by *C. gloeosporioides* hinders both ROS scavenging and flavonoid metabolism within *C. sinensis*. This may be due to either the defensive response of *C. sinensis* being triggered by the infection, disrupting the ROS scavenging and regulation of flavonoid metabolism, or the secretion of effectors by *C. gloeosporioides* into the plant cells, directly or indirectly affecting the ROS scavenging and regulation of flavonoid metabolism. CW, cell wall. PM, plasma membrane. h, haustorium. n, nuclear.

## Data Availability

Data will be made available on request. The original RNA-seq data have been uploaded to the NCBI_SRA database (PRJNA934741).

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
