# Peer review of "Colletotrichum gloeosporioides* Swiftly Manipulates the Transcriptional Regulation in *Citrus sinensis* During the Early Infection Stage"

_jof, 2024, doi:10.3390/jof10110805_

Round 1
Reviewer 1 Report
Overall, the work is well-structured, and this contribution should be considered for publication after addressing the following comments.
1. This research adds valuable knowledge to our understanding of how C. gloeosporioides infects citrus plants and circunvents their defense mechanisms. The findings can potentially lead to development new strategies for managing citrus diseases, either by breeding for resistant varieties or devising targeted treatments to bolster citrus plants' metabolic defenses. but several areas can be improved for clarity, grammar, and flow.
What are the specific mechanisms by which C. gloeosporioides suppresses CAT, POD, and SOD activities? How could these genes be manipulated to improve resistance to C. gloeosporioides? Please modify the abstract as a whole.
2. In section 2.1 modified the text why was the orchard in Xin-feng County, Jiangxi Province, China selected for this study? Are there specific environmental or agricultural factors in this region that may influence anthracnose development?
3. In section 2.2 Clarify how the conidial suspension was applied to the detached navel oranges. Was it sprayed, pipetted, or applied using another method? Please modify the text.
4. In the result discussion part section 3.1 offer more detail on why RNA degradation was observed at 24 hours post-infection. Were there specific conditions or factors that contributed to this degradation? This can help contextualize why the 24-hour samples were not used.
5. Section 4.4 Comparative Analysis: Have you compared the gene expression patterns observed in your study with those reported in other plant-pathogen interactions? Such comparisons might offer additional insights into common or unique resistance strategies, also future directions.
Overall, the work is well-structured, and this contribution should be considered for publication after addressing the following comments.
1. This research adds valuable knowledge to our understanding of how C. gloeosporioides infects citrus plants and circumvents their defense mechanisms. The findings can potentially lead to development new strategies for managing citrus diseases, either by breeding for resistant varieties or devising targeted treatments to bolster citrus plants' metabolic defenses. but several areas can be improved for clarity, grammar, and flow.
What are the specific mechanisms by which C. gloeosporioides suppresses CAT, POD, and SOD activities? How could these genes be manipulated to improve resistance to C. gloeosporioides? Please modify the abstract as a whole.
2. In section 2.1 modified the text why was the orchard in Xin-feng County, Jiangxi Province, China selected for this study? Are there specific environmental or agricultural factors in this region that may influence anthracnose development?
3. In section 2.2 Clarify how the conidial suspension was applied to the detached navel oranges. Was it sprayed, pipetted, or applied using another method? Please modify the text.
4. In the result discussion part section 3.1 offer more detail on why RNA degradation was observed at 24 hours post-infection. Were there specific conditions or factors that contributed to this degradation? This can help contextualize why the 24-hour samples were not used.
5. Section 4.4 Comparative Analysis: Have you compared the gene expression patterns observed in your study with those reported in other plant-pathogen interactions? Such comparisons might offer additional insights into common or unique resistance strategies, also future directions.
Reviewer 2 Report
Dear colleagues,
There are several questions, mainly concerning the methodology.
Line 83-93
“To reveal the effect of C. gloeosporioides infection on the dynamic expression of citrus transcriptome, probe the pathogenic mechanisms of C. gloeosporioides and concurrently screen for susceptibility genes in the genome of Citrus sinensis, we initially selected and purified a highly virulent strain of C. gloeosporioides from orchards. The strain was injected into the epicarps of C. sinensis. Samples of epicarps of C. sinensis were collected at different infection stages for assessment of physiological indicators and transcriptome sequencing. Moreover, the regulatory network of citrus fruit response to C. gloeosporioides infection was analyzed, with a detailed investigation into the expression patterns of genes related to ROS and flavonoid metabolism pathways. This study would be potential for unveiling novel insights into the interaction between citrus and C. gloeosporioides and providing substantial evidence for further dissection of the pathogenic mechanisms of C. gloeosporioides’.
There is no need to describe the methodology in such detail in the introduction; it is better to move this paragraph to the section 2. Materials and Methods
Line 96-100.
In this section, you need to separately describe the isolation of the fungal strain, its cultivation and obtaining a spore suspension.
You need to separately describe the plant varieties and their growing conditions.
Line 104-105.
«Disease assays involving C. gloeosporioides were conducted on detached fresh navel oranges by inoculating them with a conidial suspension containing 2×105 spores/mL».
Were the fruits inoculated with a needle? What volume of inoculum was used?
Line 111-112.
«A quantity of 0.5 g of citrus epicarp sample was combined with 50 mL of 70% ethanol and subjected to ultrasonic extraction for 30 minutes. Following centrifugation, the resulting supernatant was utilized for determining the flavonoid levels»
What does it mean was combined? How was the plant material ground? What was the centrifugation mode?
Line 420-421
«Furthermore, the production of ROS induced by pathogens also stimulates the synthesis of secondary metabolites in host, serving as antioxidants, to further mitigate oxidative stress»
What secondary metabolites exactly are meant? Did the authors attempt to define them?
Line 438-439
«Hence, it is hypothesized that C. gloeosporioides infection enhances its own invasion by breaking down the total flavonoids present in citrus fruits».
What is the proposed mechanism of flavonoid destruction? Can ROS be involved?
Reviewer 3 Report
The study focuses on Colletotrichum gleosporioides, a citrus disease that, in recent years, has started to damage citrus fruits. The damage is not comparable to other citrus diseases like HLB or citrus canker. However, few details are known about the molecular interaction between citrus plants and Colletotrichum gleosporioides, and an RNAseq approach can reveal candidate genes that need to be validated through knock-out and overexpression experiments. Once validated, these genes can be used for marked assisted selection (MAS) to accelerate the breeding program and identify citrus varieties resistant to the disease.
The paper identifies genes regarding ROS and flavonoid metabolic pathways during the early infection stage of Colletotrichum gleosporioides. It shows that the reduction in flavonoid content and the triggers of ROS production are key elements of this pathogen's infection and hypothesizes the presence of effector proteins that may influence ROS levels in citrus.
From Figure 1, it is unclear whether the anthracnose symptoms are evident on the fruits (it seems like something else). Did the authors continue the experiment until typical anthracnose symptoms appeared? Did the authors confirm the pathogen as C. gloeosporioides using molecular methods?
Regarding the bioinformatics analyses, why didn't the authors check the quality of the raw sequencing data? And why did they use a K-mer size of 31 with Kallisto?
It is also unclear how the authors analyzed ROS genes. Did they perform real-time PCR? Additionally, the RNA-seq data was not validated.
In the WGCNA, which genes did the authors use, and where did these 13,805 genes come from?
Round 2
Reviewer 2 Report
Dear colleagues. Thank you for your detailed answers. Good luck in your further research.
All comments have been taken into account, and the necessary information regarding the methods has been added.
Author Response
Major comments
Dear colleagues. Thank you for your detailed answers. Good luck in your further research.
[Response] We want to express our heartfelt gratitude for your time and valuable feedback. Your encouragement will indeed fuel our zeal to push the boundaries of our research. Thank you once again.
Detail comments
All comments have been taken into account, and the necessary information regarding the methods has been added.
[Response] Thank you for your valuable comments and for pointing out the areas that needed more clarity. We are pleased to confirm that we have addressed all of your comments and have added the requisite methodological details. We appreciate your continued support.
Reviewer 3 Report
Dear Authors, thank you for your responses.
I’m sorry, but based on other published articles, the symptoms of anthracnose do not seem to match. Did the authors verify this using Koch’s postulates? I have included links to two previously published articles for reference.
https://www.iris.unict.it/bitstream/20.500.11769/508902/1/agriculture-11-00536.pdf
https://www.mdpi.com/2223-7747/12/4/904#
I’m sorry, but I need more evidence to confirm that what the authors have shown are indeed anthracnose symptoms on citrus fruits. I can't proceed with an in-depth review until this question is addressed.
